# Association between chronic obstructive pulmonary disease (COPD) and occupational exposures: A hospital based quantitative cross-sectional study among the Bangladeshi population

**Ahmed Faisal Sumit** [1]*, **Anindya Das** [1], **Ishtiaq Hossain Miraj** [2], **Debasish Bhowmick** [3]

1 Department of Genetic Engineering and Biotechnology, University of Dhaka, Dhaka, Bangladesh,
2 Department of Economics, University of Dhaka, Dhaka, Bangladesh, 3 Department of Respiratory Medicine, Dhaka Medical College Hospital, Dhaka, Bangladesh

☯ These authors contributed equally to this work.
* ahmedfaisalsumit@du.ac.bd

**Data Availability Statement:** All relevant data are within the manuscript and its Supporting Information files.

## Abstract

The association between chronic obstructive pulmonary disease (COPD) and occupational exposures are less studied in Bangladeshi context, despite the fact that occupational exposures are serious public health concerns in Bangladesh. Therefore, this study aimed to evaluate this association considering demographic, health and smoking characteristics of Bangladeshi population. This was a hospital-based quantitative study including 373 participants who were assessed for COPD through spirometry testing. Assessment of occupational exposures was based on both self-reporting by respondents and ALOHA based job exposure matrix (JEM). Here, among the self-reported exposed group (n = 189), 104 participants (55%) were found with COPD compared to 23 participants (12.5%) in unexposed group (n = 184) that differed significantly (p = 0.00). Similarly, among the JEM measured low (n = 103) and high exposed group (n = 236), 23.3% and 41.5% of the participants were found with COPD respectively; compared to unexposed group (14.7%; n = 34), that differed significantly also (p = 0.00). Likewise, participants with longer self-reported occupational exposures (>8 years) showed significantly (p = 0.00) higher proportions of COPD (79.5%) compared to 40.4% in shorter exposure group (1–8 years). Similarly, significant (p = 0.00) higher cases of COPD were observed among the longer cumulative exposure years (>9 years) group than the shorter cumulative exposure years (1–9 years) group in JEM. While combining smoking and occupational exposure, the chance of developing COPD among the current, former and non-smokers of exposed group were 7.4, 7.2 and 12.7 times higher respectively than unexposed group. Furthermore, logistic analysis revealed that after adjustments for confounding risk factors, the chance of developing COPD among the self-reported exposure group was 6.3 times higher (ORs: 6.3, p = 0.00) than unexposed group; and JEM exposure group has odds of 2.8 and 1.1 respectively (p<0.05) for high and low exposures. Further studies are needed to reinforce this association between COPD and occupational exposure in Bangladesh.

**Funding:** The author(s) received no specific funding for this work.

**Competing interests:** The authors have declared that no competing interests exist.

## Introduction

COPD, a progressive lung disease characterized by airflow limitation, is largely preventable and treatable [1]. According to World Health Organization (WHO), the global prevalence of COPD in 2016 was 251 million, and around 5% of all global deaths in 2015 were attributable to COPD [2]. In South-East Asia, the COPD prevalence varied substantially ranging from 6.5% to 17.9% [3] with 8.6% in China [4] and 9% in India [5]. In Bangladesh, the prevalence of COPD was estimated around 12.5% according to the Global Initiative for Chronic Obstructive Lung Disease (GOLD) criteria [6]. Outdoor air pollutions, smoking habit, indoor air pollutions from biomass fuel burning are some of the known factors that contribute to the high prevalence of COPD in Bangladesh [3, 6].

Occupational exposure is considered to be one of the major risk factors of COPD [7, 8]. According to American Thoracic Society (ATS), around 14% of CODP cases were attributable to occupational exposures [9]. In Bangladesh, the prevalence of COPD due to occupational exposures is unknown, although occupational exposure is a serious public health concern here [10]. According to earlier report, an average of 8 million workers from all sectors suffered from workplace hazards whereas occupational exposure is identified as one of the main causes [10]. Several occupational exposures in Bangladesh had been identified, like uncontrolled pesticides exposures in farm [11], hazardous chemical exposures in tanners [12], cotton dust exposures in garments [10], biomass fuel and fumes exposures among the domestic workers [13] etc. The most alarming concern is the lack of awareness among workers, putting them into serious health threats [14].

Although, wealth of evidences had supported the association between COPD and occupational exposures worldwide [7, 8], such study has not yet been established to that extent in Bangladesh. Only COPD status has been observed among the transport workers of Dhaka city [15], and rural women who were exposed to indoor biomass fuel [16, 17]. This study, for the first time so far, attempted to determine the association between COPD and occupational exposures among the Bangladeshi population to a large extent. Here, we aimed to define occupational exposures on the basis of both self-reporting by respondents and job exposure matrix (JEM), and observe its association with COPD. We also investigated the combined effects of occupational exposures and smoking habit on COPD. Furthermore, logistic regression analysis was performed to ascertain whether these associations were influenced by other confounding factors.

## Materials and methods

### Study participants

This quantitative cross-sectional study, conducted between August, 2019 to February, 2020, included participants who were being followed-up with various respiratory symptoms in the Department of Respiratory Medicine, Dhaka Medical College Hospital, Dhaka, Bangladesh and were assessed for COPD through spirometry testing according to GOLD criteria [18]. We got 373 participants altogether based on their availability and written consent to participate. We interviewed them with the help of a physician from the department of respiratory medicine by using a structured close-ended questionnaire. The questionnaire included participants' demographic information, occupational details like types of jobs, duration of work, history of occupational exposures etc., smoking status and respiratory symptoms. The participants' weight and height were measured at the time of data collection and then BMI was calculated accordingly. The study was approved by the Ethical Review Committee of the Faculty of Biological Sciences, University of Dhaka (Ref. no. 89/Biol.Scs.).

## Spirometry testing and COPD definition

All participants underwent spirometry test with rolling seal spirometers (Sensormedics 2200, USA). COPD cases and severity were defined according to the earlier study [18]. Shortly, the participants were diagnosed with COPD when the ratio of forced expiratory volume ($FEV_1$) to forced vital capacity (FVC) had been found less than 70% (0.70) at post-bronchodilator spirometry. Post-bronchodilator spirometry was performed 10 to 15 minutes after nebulization with short-acting bronchodilator (5 mg salbutamol). The percent predicted values were estimated based on the US National Health and Nutrition Examination Survey (NHANES) III reference equation with Asian Population corrections [19]. Severity of COPD were defined as stage I (FEV1/FVC<0.70 and FEV1≥80% predicted); stage II (FEV1/FVC<0.70 and 50%≤FEV1<80% predicted); stage III (FEV1/FVC<0.70 and 30%≤FEV1<50% predicted) and stage IV (FEV1/FVC<0.70 and FEV1<30% predicted) [3].

Control participants were defined with not having COPD according to GOLD criteria (FEV1/FVC>0.70 and FEV1≥80% predicted at post-bronchodilator spirometry) other than self-reported respiratory symptoms. All control participants had been examined by registered physician and undergone both pre- and post-bronchodilator spirometry procedures. However, potential controls were excluded if they had been diagnosed with asthma, chronic bronchitis, and/or emphysema by registered physician during the time of study or possessing past history of these diseases. Furthermore, control participants with previous history of COPD or taking medications including inhalers that may affect COPD were excluded from the study. However, participants with restrictive lung disease (FEV1/FVC>0.70, and FVC<80% predicted) according to spirometry defined GOLD criteria were included in this study [20, 21].

## Evaluation of bronchodilator response

A positive bronchodilator response was evaluated based on an increase of absolute value of FEV1% ≥12% and ≥200 ml after using short-acting bronchodilator according to ATS and ERS guideline [22]. FEV1% reversibility was calculated based on the following formula: (post FEV1-Pre FEV1/ Pre FEV1) X 100 [23].

## Assessment of occupational exposure through various means

Self-reported occupational exposures were evaluated based on earlier study [24], where we asked the participants if they were exposed to vapours, gas, dust, fume, smokes, etc. in their workplace. If their self-reported responses were 'no', participants were considered as unexposed group. However, in case of positive response, participants further work details including types and duration of exposures were noted. Participants longest job status to whom they claimed exposures were considered only. The reported occupations were then coded according to the International Standard Classification of Occupations (ISCO-88) four-digit classification system [25].

Occupational exposures were also assessed using a JEM. Here, ALOHA JEM for COPD, previously used by several studies to define occupational exposures [26–28], were also established for our study. ALOHA-JEM classified all occupations, to which participants were exposed (2, 1, or 0 for 'high risk', 'low risk' or 'no' exposures, respectively), according to ISCO-88 job codes for biological and mineral dusts, gases, and fumes etc. Participants' years of exposures and category of exposures were then calculated for each job and was multiplied by that score to get cumulative exposures.

## Calculation of sample size

Sample size was calculated using this formula: $n = z^2 p (1-p)/d^2$. Considering prevalence of COPD in Bangladesh (p) = 13.5% = 0.135 according to a previous study [3], z = 1.96, margin of error (d) = 4% = 0.04; sample size (n) became 280. However, considering a large public hospital and various socio-economic classes of patients, we added 20% non-response; the sample size then rose up to 336. Then to cover the holistic dimension of our study area, we took altogether 373 samples.

## Statistical analysis

Statistical analysis was performed using SPSS program version 24 software (SPSS Inc., Chicago, USA). Univariate analysis was shown in percentage and numbers. Bivariate comparisons were done using t-test and Pearson's χ2 (chi-square) test for continuous and categorical variables respectively. The level of significance was set at $p < 0.05$. Binary logistic regression analysis was also performed to determine the adjusted associations between occupational exposures and COPD. Population Attributable Risk (%) or PAR % was determined based on the formula given by previous study [26] to find out the percentage of COPD cases attributable to occupational exposure. The calculation of PAR% was based on the following formula: PAR% = $[(Ad_{OR}-1)/Ad_{OR}]$ X Pc; where $Ad_{OR}$ = Adjusted ORs and Pc = Proportion of COPD cases exposed. The adjusted ORs were obtained from the logistic regression analysis model.

## Results

### Demographic and health characteristics of the study participants

"Table 1" depicts the demographic and health characteristics of the participants. Among the total 373 participants, 127 COPD cases (57.5% GOLD stage I and 42.2% GOLD stage II and II +) and 246 control cases (66% of total) were confirmed. The majority of the participants were aged > 60 years (52% in COPD vs 49.2% in control), male (75.6% in COPD vs 68.3% in control) and in normal BMI (70.1% in COPD vs 67.1% in control). Family history of COPD cases were found among 20.5% of COPD and 24% of control participants. Most of the participants were currently employed (70.9% in COPD vs 76.4% in control). Regarding smoking status, majority of the COPD participants were found smokers (66.9% in COPD vs 39% in control), followed by former smokers (17.3% in COPD vs 17.9% in control) and non-smokers (15.7% in COPD vs 43.1% in control). Furthermore, self-reported dyspnoea, chronic cough, morning phlegm, wheezing, and allergic rhinitis were reported by 76%, 80%, 53%, 57% and 33% of COPD participants respectively, and 60%, 49%, 44%, 47%, and 63% of control participants respectively. No significant differences ($p > 0.05$) were observed between COPD and control group regarding the participants age, gender, BMI, self-reported morning phlegm and wheezing, and self-reported family history of COPD. However, participants smoking status and self-reported dyspnoea, allergic rhinitis, and chronic cough were significantly differed ($p<0.05$) between the two groups. Besides, restrictive lung diseases had been found among 6.9% (17) of control participants ("Table 1").

### Occupational types and status among the participants

"Table 2" shows the frequency of different types of occupations reported by the participants. According to ISCO-88 four digits code, total 13 different job types were reported by the COPD and control participants. Majority of the participants of the COPD group reported their occupations as one of the following: motor vehicle mechanics (65.6%), cleaners (58.8%) and motor driver (57.6%). On the other hand, a large percentage of control reported their occupation as

**Table 1. Demographic and health characteristics of the participants.**

| Variables | GOLD Criteria | | |
|---|---|---|---|
| | CODP (n = 127) | Control (n = 246) | P value |
| **Age £** | | | |
| 40–49 years (n = 72) | 19 (15%) | 53 (21.5%) | |
| 50–59 years (n = 114) | 42 (33%) | 72 (29.3%) | 0.30 |
| >60 years (n = 187) | 66 (52%) | 121 (49.2%) | |
| **Gender** | | | |
| Male (n = 264) | 96 (75.6%) | 168 (68.3%) | 0.08 |
| Female (n = 109) | 31 (24.4%) | 78 (31.7%) | |
| **BMI $** | | | |
| Underweight (n = 88) | 31 (24.4%) | 57 (23.2%) | |
| Normal weight (n = 254) | 89 (70.1%) | 165 (67.1%) | 0.37 |
| Overweight (n = 31) | 07 (5.5%) | 24 (9.8%) | |
| **Self-reported Family history of CODP** | | | |
| No (n = 288) | 101 (79.5%) | 187 (76%) | 0.26 |
| Yes (n = 85) | 26 (20.5%) | 59 (24%) | |
| **Employment status** | | | |
| Currently Employed (n = 278) | 90 (70.9%) | 188 (76.4%) | 0.37 |
| Currently unemployed (n = 56) | 22 (17.3%) | 34 (13.8%) | |
| Retired (n = 39) | 15 (11.8%) | 24 (9.8%) | |
| **Smoking status** | | | |
| Current Smoker (n = 181) | 85 (66.9%) | 96 (39%) | |
| Former Smoker (n = 66) | 22 (17.3%) | 44 (17.9%) | 0.00 |
| Non-smoker (n = 126) | 20 (15.7%) | 106 (43.1%) | |
| **Self-reported Symptoms** | | | |
| Dyspnoea | 96 (76%) | 148 (60%) | 0.04 |
| Chronic Cough | 101 (80%) | 121 (49%) | 0.001 |
| Morning Phlegm | 67 (53%) | 108 (44%) | 0.20 |
| Wheezing | 72 (57%) | 116 (47%) | 0.32 |
| Allergic rhinitis | 42 (33%) | 156 (63%) | 0.001 |
| **Presence of restrictive disease** | | | |
| Yes | | 17 (6.9%) | |
| No | | 229 (93.1%) | |
| **COPD stages (GOLD Criteria)** | | | |
| Stage I | 73 (57.5%) | | |
| Stage II and II+ | 54 (42.5%) | | |

$ BMI was categorized into underweight (BMI <18.5), normal weight (18.5–25) and overweight (>25) according to WHO classification [29].

£ Age was categorized based on earlier study that determined COPD prevalence in Bangladesh [3].

one of the following: manager (92%), clerk (90%), house-keeper (90%), administrative job (85.7%), accountant (83.3%), builder (75%), farmer (68.1%), and salesperson (65.5%).

## Occupational exposure significantly affected COPD

We found that COPD cases were observed among 55% of self-reported exposed and 12.5% of self-reported unexposed participants ("Table 3"). On the other hand, control cases were found among 45% of self-reported exposed and 87.5% of unexposed participants. While, according

**Table 2. Types of jobs reported by the participants.**

| Type of Jobs | GOLD Criteria | | |
|---|---|---|---|
| | COPD (n = 127) | Control (n = 246) | P value |
| Motor vehicle Mechanic (n = 32) | 21 (65.6%) | 11 (34.4%) | 0.00 |
| Tannery Worker (n = 5) | 02 (40%) | 03 (60%) | |
| Cleaner (n = 51) | 30 (58.8%) | 21 (41.2%) | |
| Motor driver (n = 33) | 19 (57.6%) | 14 (42.4%) | |
| Manager (n = 25) | 02 (08%) | 23 (92%) | |
| Clerk (n = 20) | 02 (10%) | 18 (90%) | |
| Garment worker (n = 20) | 06 (30%) | 14 (70%) | |
| Housekeepers and related worker (n = 40) | 04 (10%) | 36 (90%) | |
| Farmer (n = 47) | 15 (31.9%) | 32 (68.1%) | |
| Salesperson (n = 58) | 19 (32.7%) | 39 (67.3%) | |
| Administrative Professional (n = 28) | 04 (14.3%) | 24 (85.7%) | |
| Accountant (n = 06) | 01 (16.7%) | 05 (83.3%) | |
| Builder (n = 08) | 02 (25%) | 06 (75%) | |

to JEM exposure measurement, only 14.7% of the participants with unexposed category developed COPD, whereas the proportion increased to 23.3% and 41.5% for low and high-risk job categories respectively. All these values in COPD differed significantly (p = 0.00) than control group ("Table 3").

The duration of exposure (years) among the self-reported exposure group differed significantly as well between the two groups. The mean value of the exposure years for COPD group were 11.10 ± 5.52 years, whereas that for the control group were 5.19 ± 3.58 years. On the other hand, the mean value of JEM cumulative exposure years was around 3 times higher for the COPD group (19.69 ± 11.86 years) compared to the control group (6.94 ± 5.92 years), and that differed significantly (p = 0.00) as well ("Table 3").

## Longer duration of occupational exposure significantly related to COPD

The median value of the exposure years was used as a cut-off point to divide the duration of exposure years into two sub-groups: shorter exposure and longer exposure years. In case of self-reported exposure years, the median value of the exposure years among the 189 self-reported exposure participants was found 8. On the other hand, the median value for the

**Table 3. Occupational exposures among the COPD and control group.**

| Exposure measurement | GOLD criteria | | |
|---|---|---|---|
| | COPD (n = 127) | Control (n = 246) | P value |
| **Self-reported exposure measurement** | | | 0.00 |
| Unexposed (n = 184) | 23 (12.5%) | 161 (87.5%) | |
| Exposed (n = 189) | 104 (55%) | 85 (45%) | |
| ^Self-reported exposure-years (Mean ± SD) | 11.10 ± 5.52 | 5.19 ± 3.58 | 0.00 |
| **JEM exposure Measurement** | | | 0.00 |
| Unexposed (n = 34) | 05 (14.7%) | 29 (85.3%) | |
| Low risk (n = 103) | 24 (23.3%) | 79 (76.7%) | |
| High risk (n = 236) | 98 (41.5%) | 138 (58.5%) | |
| ^JEM Cumulative exposures-years (Mean ± SD) | 19.69 ± 11.86 | 6.94 ± 5.92 | 0.00 |

^ Self-reported exposure years and JEM cumulative exposure years (Unit, years) were restricted to exposed participants only.

**Table 4. Influence of longer and shorter occupational exposure years (both cumulative and self-reported) on COPD.**

| | GOLD criteria | | | |
|---|---|---|---|---|
| | COPD | Control | P value | ORs |
| **Cumalative Exposure Years**[*] | | | | |
| Shorter Exposure Years (n = 170) (1–9 years) | 35 (20.6%) | 135 (79.4%) | 0.00 | 4.09 |
| Longer Exposure Years (n = 169) (>9 years) | 87 (51.5%) | 82 (48.5%) | | |
| **Self-reported expossure years** | | | | |
| Shorter Exposure Years (n = 111) (1–8 years) | 42 (40.4%) | 69 (59.6%) | 0.00 | 12.44 |
| Longer Exposure Years (n = 78) (>8 years) | 62 (79.5%) | 16 (20.5%) | | |

[*] Cumulative exposure year was the multiplicative product of years of exposures and category of exposures (2, 1, or 0 for 'high risk', 'low risk' or 'no' exposures, respectively) defined by ALOHA JEM.

weighted cumulative exposure years (according to JEM exposure probability) among the 339 participants was found 9.

Here, in case of JEM cumulative exposure years, we found that 20.6% of participants with shorter duration (1–9 years) developed COPD compared to 51.5% participants with longer duration (>9 years) (ORs: 4.09, p = 0.00) ("Table 4"). While, in case of self-reported exposure years, 79.5% of participants with longer duration (>8 years) developed COPD compared to 40.4% of participants with shorter duration (1–8 years) (ORs: 12.44, p = 0.00) ("Table 4").

## Occupationally exposed COPD participants didn't show any significance in positive bronchodilator response compared to the unexposed group

Since this study excluded control participants with positive bronchodilator response due to the suspect of asthma, the comparisons were performed among the occupationally exposed and unexposed COPD participants only ("Table 5"). Here, we found that among the self-reported unexposed group, the percentage of participants with positive bronchodilator response were slightly higher (17.4%) than that of the self-reported exposed group (15.4%), but not significantly differed (p = 0.51). Similarly, according to JEM exposure measurement, no significant differences (p = 0.86) were observed among the unexposed, low-risk and high-risk exposed COPD participants in terms of positive bronchodilator response.

## Smoking habit significantly increased the rate of COPD among the occupationally exposed group

The combined influence of smoking habit and self-reported occupational exposures on COPD were shown in "Table 6". Here, 65.1% COPD cases were observed among the participants with

**Table 5. Bronchodilator response among the COPD participants.**

| Exposure measurement | FEV1 Reversibility among the COPD participants | | P value |
|---|---|---|---|
| | (≥ 12% and ≥ 200 ml) | <12% and <200ml | |
| **Self-reported** | | | |
| Unexposed (n = 23) | 04 (17.4%) | 19 (82.6%) | 0.51 |
| Exposed (n = 104) | 16 (15.4%) | 88 (84.6%) | |
| **JEM exposure measurement** | | | |
| Unexposed (n = 05) | 01 (20%) | 04 (80%) | 0.86 |
| Low risk (n = 24) | 03 (12.5%) | 21 (87.5%) | |
| High risk (n = 98) | 16 (16.3%) | 82 (83.7%) | |

**Table 6. Combined influence of smoking habit and self-reported occupational exposures on COPD.**

| Exposure and smoking status | GOLD criteria | | | |
|---|---|---|---|---|
| | COPD (n = 127) | Control (n = 246) | P value | ORs |
| Current smoker and Exposed group (n = 106) | 69 (65.1%) | 37 (34.9%) | 0.000 | 7.4 |
| Current smoker and Unexposed group (n = 74) | 15 (20.3%) | 59 (79.7%) | | |
| Former smoker and Exposed group (n = 31) | 17 (54.8%) | 14 (45.2%) | 0.001 | 7.2 |
| Former smoker and Unexposed group (n = 35) | 05 (14.3) % | 30 (85.7%) | | |
| Non-smoker and Exposed group (n = 52) | 18 (34.6%) | 34 (51.5%) | 0.000 | 12.7 |
| Non-smoker and Unexposed group (n = 75) | 03 (4%) | 72 (96%) | | |

self-reported occupational exposures and current smoking habit, followed by the former smokers (54.8% COPD cases) and non-smokers (34.6% COPD cases). While, 20.3% COPD cases were observed among the self-reported unexposed participants with current smoking habit, whereas that value for the former smokers and non-smokers were 14.3% and 4% respectively. The chance of developing COPD among the current, former and non-smokers of occupationally exposed group were 7.4, 7.2 and 12.7 times higher respectively than those smoking categories of unexposed group. All the differences were significant as well (p<0.05) ("Table 6").

## Binary logistic regression analysis

Binary logistic regression analysis was performed considering the presence or absence of COPD as a dependent variable. We found that after adjustments with age, gender, BMI, smoking habit, family history of COPD and cumulative exposure years, the risk of COPD among self-reported exposure group was 6.3 times higher than unexposed group (p = 0.00) ("Table 7"). After considering JEM defined cumulative exposure years, the odds of having

**Table 7. Binary logistic regression analysis to associate COPD with other independent variables and PAR%.**

| Variables | Category of Characteristics | Adjusted ORs (95% CI) | P-value | PAR% |
|---|---|---|---|---|
| **Age** | 40–49 years (Reference) | 1 | | |
| | 50–59 years | 1.4 (0.58–2.9) | 0.51 | 9.4 |
| | >60 years | 1.6 (0.71–3.23) | 0.28 | 19.5 |
| **Gender** | Male (Reference) | 1 | | |
| | Female | 1.6 (0.82–4.18) | 0.13 | 28.3 |
| **BMI** | Normal (Reference) | 1 | | |
| | Underweight | 1.6 (0.39–1.46) | 0.10 | 9.15 |
| | Overweight | 0.6 (0.22–1.85) | 0.41 | 3.7 |
| **Smoking habit** | Never (Reference) | 1 | | |
| | Current | 5.0 (2.2–12.1) | 0.00 | 53.5 |
| | Former | 2.8 (1.1–7.4) | 0.02 | 11.1 |
| **Family history of COPD** | Yes (Reference) | 1 | | |
| | No | 1.0 (0.51–1.96) | 0.99 | 1.8 |
| **Self-reported exposure** | Unexposed (Reference) | 1 | | |
| | Exposed | 6.3 (2.8–9.2) | 0.00 | 46.2 |
| **Exposure Duration (Cumulative years)** | Unexposed (Reference) | 1 | | |
| | Shorter exposure years | 1.1 (1.0–12.08) | 0.7 | 2.1 |
| | Longer exposure years | 2.8 (1.2–13.09) | 0.05 | 26.7 |

Abbreviation: CI: Confidence Interval, ORs: Odds ratio.

COPD among the high and low- exposure years group were 2.8 and 1.1 respectively (p<0.05), compared to unexposed group while adjusting other variables (p<0.05).

Smoking habit also showed positive significant association (Adjusted ORs = 5.0 and 2.8, p<0.05 for current and former smokers respectively) with COPD when compared to non-smokers. Similarly, gender (female; ORs = 1.6), BMI (underweight; ORs = 1.6), and age (50–59 years; ORs = 1.4 and >60 years; ORs = 1.6) showed positive associations with COPD, but these associations were not found significant (p>0.05) ("Table 7").

We further calculated PAR% to assess the proportion of the incidence of COPD in Bangladeshi population due to occupational exposure. Here, PAR% for COPD was found 46.2 for self-reported exposure group and 2.1 and 26.7 for JEM defined low and high-exposure years group respectively. For current and former smokers, PAR% for COPD were found 53.5 and 11.1 respectively.

## Discussion

This is the first kind of hospital-based quantitative cross-sectional study that determined the association between COPD and occupational exposures among the Bangladeshi population. Here, it is observed that among the total 373 participants, 127 participants (34% of total) were found suffering from spirometry defined COPD. Depending on the assessment measures, COPD cases had been observed among 55%, 41.5% and 23.3% participants with self-reported occupational exposures, JEM exposure defined high and low risk occupation categories respectively. The presence of COPD- occupational exposure relationship was further confirmed after adjusting with cofounding risk factors.

Our results supported other literatures regarding the significant association between COPD and occupational exposures [7, 8]. We got almost 6-folds increased adjusted ORs for self-reported occupational exposures compared to unexposed group. These findings were in accord with previous studies that reported adjusted ORs ranging from 1.3 to 5.9 for COPD due to occupational exposures [9, 26]. Furthermore, the PAR% for COPD among the self-reported, low and high-cumulative exposure years group were found 46.2%, 26.7% and 2.1%, respectively. However, these were slightly higher compared to previous study [26] that reported PAR% for COPD ranging from 25% to 2.3% for self-reported and JEM measured exposures to variable gases, chemicals and dusts.

Our study also investigated combined influence of occupational exposures and smoking on COPD. Our unadjusted data showed that joint effect of occupational exposure and current smoking habit was associated with nearly 7-folds increase in case of developing COPD, similar to previous study that showed nearly 5-folds increase in the risk of developing COPD [30], but slightly lower than another study that showed nearly 14-folds of increase [9]. However, for adjusted data, it was observed that current and former smoking habit were associated with 5.0 and 2.8-times higher risks of developing COPD respectively than non-smokers, which was almost consistent with previous study [31] that showed 4.40-times of higher risks. The PAR% for COPD due to current smoking habit was found 53.5%, which supported previous data that showed 56% for combined current and past smoking habit [32].

The duration of occupational exposures required to develop COPD is still in ambiguity. According to earlier report, minimum 15 years of exposures were required to develop COPD [33], whereas another study showed 6–10 years of exposures [27]. Here, we defined low and high exposure years based on the previous study where median value of the cumulative exposure years was used as cut off point to categorize exposure duration [28]. Here, our adjusted data revealed that longer and shorter duration of cumulative exposure years increased the risk of COPD by 2.9 and 1.2-folds respectively compared to unexposed group which was also

correlated with pervious study that showed 1.3 to 2.2-folds increase of COPD due to low and high cumulative exposure years respectively [26].

The association between occupational exposure and bronchodilator response among the COPD participants were also assessed in our study. Here, similar to earlier study [34], our study showed that 12–20% of COPD participants exhibited positive bronchodilator response, although the association was not significant when compared to occupational exposure. Our finding was in contrast with previous study showing the significant association between % FEV1 reversibility with occupational exposure among the COPD participants [23]. However, such assessment in the previous study [23] was performed only based on the increase of average %FEV1 predicted values among the exposed group rather than on positive bronchodilator response.

We, similar to other studies [3, 35, 36], found correlations between COPD and low BMI (Adjusted ORs: 1.6); and between COPD and age (Adjusted ORs: 1.6 for age>60 years and1.4 for age 50–59 years), although those were not significant. Furthermore, similar to previous study [28], we also found positive association between COPD and gender (female). However, unlike other studies [37, 38], we did not find positive association between COPD and family history of COPD. This might happen because we noted self-reported response without checking authenticity, and most of our participants were not educated enough.

Our study had few limitations. This was hospital-based study that might not represent the global scenario of Bangladesh. Many of the variables including smoking habit, age and family history of COPD were self-reported without testing their authenticity. We could not verify occupational exposures reported by the participants. However, this limitation was addressed by checking participants job identity card and moreover by conducting JEM exposure measures. Besides, exposure assessment from longer job durations in our study helped to ensure that occupational exposure led to the development of COPD. Another limitation of our study was to define COPD according to GOLD criteria only which was reported previously to overestimate COPD cases [3]. This was also reflected in our study as well since both prevalence and ORs of developing COPD for most variables were found higher from the similar studies [9, 26].

Our study guided us to conclude that occupational exposures in the workplace was significantly associated with COPD among Bangladeshi population. Additional factors including smoking habit, age, BMI and gender should also be kept in mind as those became important risk factors of COPD too. Further research is needed to look over this association into more detail to mitigate COPD cases among Bangladeshi people.

## Supporting information

**S1 Table. Spirometry values of control participants according to self-reported occupational exposure.**
(PDF)

**S2 Table. Spirometry values of COPD participants according to self-reported occupational exposure.**
(PDF)

## Acknowledgments

We are thankful for the support and help from the staff and attendants of the Dhaka Medical College Hospital, Dhaka, Bangladesh. Furthermore, we are grateful to the participants and their family who have agreed either to participate or give consent in the study.

## Author Contributions

**Conceptualization:** Ahmed Faisal Sumit, Anindya Das.

**Formal analysis:** Ahmed Faisal Sumit, Anindya Das.

**Investigation:** Ahmed Faisal Sumit, Anindya Das, Ishtiaq Hossain Miraj, Debasish Bhowmick.

**Methodology:** Ahmed Faisal Sumit, Anindya Das, Ishtiaq Hossain Miraj.

**Project administration:** Ahmed Faisal Sumit.

**Supervision:** Ahmed Faisal Sumit, Anindya Das.

**Validation:** Ahmed Faisal Sumit.

**Writing – original draft:** Ahmed Faisal Sumit.

**Writing – review & editing:** Ahmed Faisal Sumit, Anindya Das.

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
