## [Decision Letter · Decision Letter 0]

16 Jul 2020

PONE-D-20-16525

Association between chronic obstructive pulmonary disease (COPD) and occupational exposures: A hospital based quantitative cross-sectional study among the Bangladeshi population.

PLOS ONE

Dear Dr. Sumit,

Thank you for submitting your manuscript to PLOS ONE. After careful consideration, we feel that it has merit but does not fully meet PLOS ONE’s publication criteria as it currently stands. Therefore, we invite you to submit a revised version of the manuscript that addresses the points raised during the review process.

ACADEMIC EDITOR: Please insert comments here and delete this placeholder text when finished. Be sure to:

Indicate which changes you require for acceptance versus which changes you recommendAddress any conflicts between the reviews so that it's clear which advice the authors should followProvide specific feedback from your evaluation of the manuscript

We look forward to receiving your revised manuscript.

Kind regards,

Christophe Leroyer

Academic Editor

PLOS ONE

**Comments to the Author**

1. Is the manuscript technically sound, and do the data support the conclusions?

Reviewer #1: Partly

Reviewer #2: Yes

2. Has the statistical analysis been performed appropriately and rigorously? 

Reviewer #1: Yes

Reviewer #2: Yes

3. Have the authors made all data underlying the findings in their manuscript fully available?

Reviewer #1: Yes

Reviewer #2: Yes

4. Is the manuscript presented in an intelligible fashion and written in standard English?

Reviewer #1: Yes

Reviewer #2: Yes

5. Review Comments to the Author

Reviewer #1: The authors examine the relation between occupational exposures and COPD in a respiratory clinic based population in Bangladesh. They find a higher rate of COPD in Occupationally exposed population.

Major comments:

As all of the patients were being evaluated for respiratory problems, the authors need to be very cautious in how these results are presented and interpreted. For example- what diagnoses did the patients in the control group actually have?

Also- the authors should present the proportion of patients with restriction ( normal ratio and low FEV1)- As I suspect there would be a high proportion of patients with this abnormality.

Was pre bronchodilator lung function assessed? This might be important and including an assessment of bronchodilator responsiveness would be helpful.

Reviewer #2: The authors present original results concerning the share of professional origin in the etiology of COPD in Bangladesh. The text is well written apart from 2 minimal mistakes. The document is understandable and meets the criteria of a good scientific article. The publication is original and will allow to deepen later among the trades found, the pathogens at the origin of these respiratory diseases.

6. PLOS authors have the option to publish the peer review history of their article (what does this mean?). If published, this will include your full peer review and any attached files.

Reviewer #1: **Yes: **David Mannino

Reviewer #2: **Yes: **DEWITTE, J.D.

---

## [Author Response · Author response to Decision Letter 0]

22 Aug 2020

Dear Respected Reviewers,

Many thanks for reviewing our paper. We really appreciate that you gave us response in such a short time. We have edited the manuscript according to your suggestions. We are grateful enough to receive the inputs you both have given for our manuscript. We strongly believe that your inputs will help us improving the quality of our manuscript substantially. 

Please find the responses here:

Response to Reviewers' Comments

Reviewer 1 comment:

1. As all of the patients were being evaluated for respiratory problems, the authors need to be very cautious in how these results are presented and interpreted. For example- what diagnoses did the patients in the control group actually have?

Response: Initially, all the control participants were examined by registered physician after complaining about respiratory problems. They all were confirmly diagnosed without COPD based on the spirometry results that was FEV1/FVC>0.70, and FEV1% ≥ 80 predicted.

Based on the suggestions by registered physician, some of the control participants were also undergoing other tests like chest x-ray, serum IgE test, blood eosinophil count, etc, to confirm the diagnosis of asthma, bronchitis, emphysema, restrictive problems and/or other respiratory diseases. We have excluded those control participants who had been diagnosed with asthma, chronic bronchitis, and/or emphysema by registered physician during the time of study or if they had past history of these diseases. This information has been added in the manuscript now.

We have not excluded control participants if they had been diagnosed with restrictive problems. Besides, control participants with doctor diagnosed allergic rhinitis, wheezing, mild/chronic cough, morning phlegm, dyspnoea were also included in our study. However, majority of the control participants were diagnosed as normal by registered physician.

In our study we have only assessed self-reported respiratory symptoms of the participants (“Table 1”) rather than registered physician diagnosed respiratory diseases since we found majority of the control participants were diagnosed as normal by registered physician.

2. Also- the authors should present the proportion of patients with restriction (normal ratio and low FEV1)- As I suspect there would be a high proportion of patients with this abnormality.

Response: According to your suggestion, we have included the proportion of control participants with restriction (normal ratio, but FVC<80% predicted) according to GOLD criteria.

3. Was pre bronchodilator lung function assessed? This might be important and including an assessment of bronchodilator responsiveness would be helpful.

Response: According to your suggestion, we have included pre bronchodilator response data in our manuscript.

Since we excluded all control participants who have shown positive bronchodilator response due to the suspect of asthma, bronchodilator response was shown only for the COPD participants.

Reviewer 2 comment:

The text is well written apart from 2 minimal mistakes. 

Response: We have addressed short grammatical mistakes and corrected accordingly.

---

## [Editor Report · Decision Letter 1]

3 Sep 2020

PONE-D-20-16525R1

Association between chronic obstructive pulmonary disease (COPD) and occupational exposures: A hospital based quantitative cross-sectional study among the Bangladeshi population.

PLOS ONE

Dear Dr. Sumit,

Thank you for submitting your manuscript to PLOS ONE. After careful consideration, we feel that it has merit but does not fully meet PLOS ONE’s publication criteria as it currently stands. Therefore, we invite you to submit a revised version of the manuscript that addresses the points raised during the review process.

ACADEMIC EDITOR:

My comment refers to the discussion section:

"The association between occupational exposure and bronchodilator response among the COPD

300 participants were also assessed in our study. Here, similar to earlier study [34], our study showed that

301 12-20% of COPD participants exhibited positive bronchodilator response, although the association was

302 not significant when compared to occupational exposure. Our finding was in contrast with previous

303 study showing the significant association between %FEV1 reversibility with occupational exposure

304 among the COPD participants [23]. However, such assessment was performed only based on the

305 increase of average %FEV1 predicted values among the exposed group rather than on positive

306 bronchodilator response."

The last sentence is unclear to me: do you refer to your study or to the previous one ?

many thnaks to elucidate this point

We look forward to receiving your revised manuscript.

Kind regards,

Christophe Leroyer

Academic Editor

PLOS ONE

---

## [Author Response · Author response to Decision Letter 1]

9 Sep 2020

Reviewer comment:

The association between occupational exposure and bronchodilator response among the COPD participants were also assessed in our study. Here, similar to earlier study [34], our study showed that 12-20% of COPD participants exhibited positive bronchodilator response, although the association was not significant when compared to occupational exposure. Our finding was in contrast with previous study showing the significant association between %FEV1 reversibility with occupational exposure among the COPD participants [23]. However, such assessment was performed only based on the increase of average %FEV1 predicted values among the exposed group rather than on positive bronchodilator response."

The last sentence is unclear to me: do you refer to your study or to the previous one?

Response:

The last sentence refers to previous study and we have added this in our manuscript as follow:

“However, such assessment in the previous study [23] was performed only based on the increase of average %FEV1 predicted values among the exposed group rather than on positive bronchodilator response.”

---

## [Editor Report · Decision Letter 2]

10 Sep 2020

Association between chronic obstructive pulmonary disease (COPD) and occupational exposures: A hospital based quantitative cross-sectional study among the Bangladeshi population.

PONE-D-20-16525R2

Dear Dr. Sumit,

We’re pleased to inform you that your manuscript has been judged scientifically suitable for publication and will be formally accepted for publication once it meets all outstanding technical requirements.

Kind regards,

Christophe Leroyer

Academic Editor

PLOS ONE

---

## [Editor Report · Acceptance letter]

14 Sep 2020

PONE-D-20-16525R2 

Association between chronic obstructive pulmonary disease (COPD) and occupational exposures: A hospital based quantitative cross-sectional study among the Bangladeshi population. 

Dear Dr. Sumit:

I'm pleased to inform you that your manuscript has been deemed suitable for publication in PLOS ONE. Congratulations! Your manuscript is now with our production department. 

Kind regards, 

on behalf of

Dr. Christophe Leroyer 

Academic Editor

PLOS ONE